# Correlation between UV Index, Temperature and Humidity with Respect to Incidence and Severity of COVID 19 in Spain

**DOI:** 10.3390/ijerph20031973

**Published:** 2023-01-20

**Authors:** Juan Blas Pérez-Gilaberte, Natalia Martín-Iranzo, José Aguilera, Manuel Almenara-Blasco, María Victoria de Gálvez, Yolanda Gilaberte

**Affiliations:** 1Department of Internal Medicine, Miguel Servet University Hospital, IIS Aragon, 50009 Zaragoza, Spain; 2Medical School, University of Zaragoza, 50009 Zaragoza, Spain; 3Photobiological Dermatology Laboratory Medical Research Center, Department of Dermatology and Medicine, School of Medicine, Campus Universitario de Teatinos S/N, 29071 Málaga, Spain; 4Department of Dermatology, Miguel Servet University Hospital, IIS Aragon, 50009 Zaragoza, Spain

**Keywords:** COVID-19 incidence, hospitalizations, mortality, ultraviolet radiation, temperature

## Abstract

Background: Various studies support the inverse correlation between solar exposure and Coronavirus SARS-CoV-2 infection. In Spain, from the Canary Islands to the northern part of the country, the global incidence of COVID-19 is different depending on latitude, which could be related to different meteorological conditions such as temperature, humidity, and ultraviolet index (UVI). The objective of the present work was to analyze the association between UVI, other relevant environmental factors such as temperature and humidity, and the incidence, severity, and mortality of COVID-19 at different latitudes in Spain. Methods: An observational prospective study was conducted, recording the numbers of new cases, hospitalizations, patients in critical units, mortality rates, and annual variations related to UVI, temperature, and humidity in five different provinces of Spain from January 2020 to February 2021. Results: Statistically significant inverse correlations (Spearman coefficients) were observed between UVI, temperature, annual changes, and the incidence of COVID-19 cases at almost all latitudes. Conclusion: Higher ultraviolet radiation levels and mean temperatures could contribute to reducing COVID-19 incidence, hospitalizations, and mortality.

## 1. Introduction

The SARS-CoV-2 coronavirus pandemic is believed to have begun in December 2019 in Wuhan, China [1]. A cough, fever, or sore throat are some of the most common symptoms in adolescent and adult patients. However, severe cases may present ARDS (adults respiratory distress syndrome) with refractory hypoxemia, multiple organ failure, and even death; most of these cases require conventional hospitalization and even intensive care unit (ICU) treatment [2]. Diagnosis suspicion is clinical at first, based on the symptoms that the patient develops. Microbiological diagnosis is mandatory, by PCR test on nasopharyngeal smear or serological testing, in order to confirm an active infection. Since the pandemic started, more than 4,900,000 people have died from COVID-19 disease across the world, most of whom were over 65 years old [3].

The virus is transmitted mainly by exposure to respiratory fluids from an infected person. The three main methods of exposure are: (1) inhalation of fine droplets and aerosol particles, (2) respiratory droplets and particles making contact with exposed mucous membranes, including by direct splashes or sprays, and (3) mucous membranes coming in contact with hands previously soiled by exhaled respiratory fluids containing the virus, or from touching inanimate surfaces contaminated with the virus [4,5]. Generally, respiratory disease-causing viruses are more contagious when the infected person is symptomatic. Basic hygienic rules to prevent transmission include hand washing, social distancing, mask wearing, and coughing or sneezing into disposable tissues.

Environmental factors, especially solar ultraviolet (UV) radiation, have been reported to play a determinant role in outdoor infection patterns. Around 99% of solar radiation reaching earth’s surface ranges between 0.2 and 3.0 µm, from UV to infrared radiation. UV radiation varies throughout the year, showing greater levels during spring and summer, and lower in autumn and winter [6]. UV radiation exerts different effects on human health and the environment. Regarding health, UVB radiation is crucial for vitamin D synthesis in the skin, and consequently in regulating all the functions of vitamin D, including its immune modulatory effect; several studies support the association between low serum vitamin D levels and more severe COVID-19 disease [7,8]. In regard to the environmental effect of UV radiation, UVC has been shown to be very effective in decontaminating the air from SARS-CoV-2 [9]. Solar UV radiation reaching earth’s surface does not contain UVC, due to its absorption by atmospheric oxygen (wavelengths below 200 nm) and the stratospheric ozone layer (wavelengths of 200–290 nm). However, solar UVB as well as UVA radiation reaching the earth’s surface has been proposed to be an environmental determinant shaping COVID-19 transmission at the seasonal timescale [10] and has been reported to be clearly negatively correlated with infection levels and mortality of patients [11,12,13,14]. Moreover, wavelengths other than solar UVB have been demonstrated to be involved in COVID-19 inactivation. Nicastro et al. [15] and Biasin et al. [16] published the action spectrum of COVID-19 inactivation, and reported that wavelengths of 366 and 405 nm are effective for that inactivation. The authors indicated that SARS-CoV-2 as well as other RNA viruses are particularly sensitive to solar UV radiation. Doses of UVA necessary for virus inactivation were 10^4^ times higher than UVC, although data confirm that summer UV exposure could provide sufficient UVB and UVA doses for virus inactivation in less than 2 min outdoors.

The ultraviolet index (UVI) measures UV radiation that potentially causes skin erythema, such as is found during the daytime and at the earth’s surface [17]. Because UVB radiation is the main part of UV solar spectrum that promotes skin erythema, and is also related to DNA damage due to similar action spectra [18,19,20], UVI is considered the main indicator for the potential solar damage to humans. In this context, its use has been globally standardized by almost every weather agency [21]. Various researchers have described seasonality of infection patterns, as well as reporting the inactivation of the SARS-CoV-2 virus as a function of standard UVI [22,23,24]. A standardized scale is used globally to measure UVI according to different grades of exposure: 0–2 (low exposure), 3–5 (mild exposure), 6–7 (high exposure), 8–10 (very high exposure), and 11 or more (extremely high exposure). Other climate factors such as the thickness of the ozone layer and the presence of clouds or air pollution can introduce changes in UV radiation reaching the earth’s surface [24].

Temperature seems also to be important for SARS-CoV-2, which has been reported to be highly stable at 4 °C but sensitive to heat [11]. Several studies suggest that environmental conditions may have affected the COVID-19 pandemic, although the results are controversial. In this respect, certain temperature and humidity increases could partially suppress outdoor transmission of COVID-19 [25,26].

Taking all of this into consideration, our objective was to analyze the associations between UVI, other relevant environmental factors such as temperature and humidity, and the incidence, severity, and mortality of COVID-19 at different latitudes in Spain.

## 2. Material and Methods

A prospective observational study was conducted, selecting incident cases, hospitalized cases, ICU-admitted cases, and deaths caused by COVID-19 infection from January 2020 to February 2021 in five provinces in Spain. The provinces were selected to cover the whole latitude of Spain from north to south, as follows: Guipuzcoa (latitude N 43.257°), Zaragoza (N 41.6563°), Madrid (N 40.4167°), Málaga (N 36.72016°), and Tenerife (N 28.46824°). In order to compare the different cities, we obtained the incidence rates of the four variables per 100,000 inhabitants, according to the number of cases and the total population of each province: Madrid 6,779,888 inhabitants, Málaga 1,685,929, Tenerife 1,044,887, Zaragoza 972,528, and Guipuzcoa 727,212. The data were collected from the National Epidemiology Centre (CNE) at the Official Carlos III Institute of Health. The Institute provides daily detailed information on the demographic, epidemiological, and clinical characteristics of COVID-19 patients. The data are obtained from individualized information from different Spanish Health Centers and are available to the general public. Data are anonymously sourced and can be used for statistical analysis without further requirements. Data were obtained from https://cnecovid.isciii.es/ (accessed on 14 March 2021), and no ethical approval was necessary to use the data for this work. Meteorological data in terms of daily maximal UVI, mean humidity, and temperature for the five locations were collected from the National Meteorology Agency in Spain (AEMET). Daily data were collected from CNE and AEMET, and are represented in figures by mean data per week, taking information for the first, second, and third weeks from days 1–7, 8–14, and 15–21, respectively. Data for the fourth week were the mean values from day 22 to the end of each month. For comparison between the different latitudes, mean data per season are also represented, as well as the mean of the total study period (Table 1).

As the variables did not follow normality, Spearman correlation testing was applied to analyze the correlations between daily meteorologic variables and incidental cases of the different variables of COVID-19. The interpretation level of the correlation coefficient for the different variables analyzed was divided into 0.1–0.39 as weak correlation, 0.4–0.69 as moderate, 0.7–0.89 as strong, and 0.9–1 as very strong correlation [27].

COVID-19 infection is sometimes diagnosed several days after the patient becomes infected, with a mean incubation period of five days [28]. With the aim of evaluating the influence of weather on COVID-19 transmission and severity, in order to perform an accurate analysis, we assessed the correlation of COVID-19 incidence with meteorological data from five days previously. This method was also applied for hospitalizations, correlating with meteorological data from nine days earlier [29], for ICU admissions, correlating with data from fifteen days earlier, and deaths, which were correlated with data from twenty days earlier [28].

## 3. Results

### 3.1. Description of Meteorological and COVID-19 Variables

Meteorological data (UVI, temperature, and humidity) and weekly changes in the incidence of COVID 19 (total cases and hospitalizations/100,000 inhabitants) are represented in Figure 1. Seasonal data for all meteorological and incidence variables are described in Table 2. Annual variations in UVI were observed, from lower values (around 1) at the beginning of January to higher values in July (around 10). The exception was Tenerife, with variations of UVI from lower winter values of around 3 to maximal values of 12 at the beginning of July. The mean UVI ratings per season are represented in Table 1, showing significant variations from San Sebastian values with respect to other southern locations. Similar annual patterns were observed for temperature and humidity in the peninsula provinces, with a gradual increase of temperature from winter to mid-summer and the inverse observed for humidity. The exception was Tenerife with lower annual variations of temperature around 25 °C, and humidity over 70% for the whole year.

The highest mean value for the incidence of COVID-19 cases per 100,000 inhabitants per day over the whole study period was observed in Madrid (21.39, SD 15.8), followed by Zaragoza (18.89, SD 15.3), Guipuzcoa (16.73, SD 16.9). The rate of incidence in Málaga was almost the half that of Madrid (12.84, SD 15.7) and the lowest rate was in Tenerife (4.05, SD 4.0). Differences between Tenerife and the other provinces were all statistically significant (*p* < 0.001).

The mean number of daily hospitalizations per 100,000 inhabitants was also lower in Tenerife (1.02, SD 1.3) compared with the rest of the studied cities, with statistically significant differences when compared with all other provinces (*p* < 0.05) (Table 1). The mean number of ICU-admitted patients per 100,000 inhabitants per week was significantly higher in Zaragoza (0.2, SD 0.1), followed by Madrid (0.16, SD 0.11), compared with the rest of the provinces. Differences between Zaragoza and Guipuzcoa (*p* = 0.001), Málaga (*p* = 0.002), and Tenerife (*p* = 0.006) were statistically significant. Higher mean values for deaths were seen in Zaragoza (0.56, SD 0.33) and Madrid (0.46, SD 0.27), followed by Guipuzcoa (0.39, SD 0.29), Málaga (0.22, SD 0.09), and Tenerife (0.08, SD 0.05). Differences in the mean values for deaths between Málaga and Tenerife on one hand and the northern provinces on the other were statistically significant (*p* < 0.05).

### 3.2. Correlation between COVID-19 and UVI

A statistically significant inverse correlation was observed between UVI and the numbers of daily incidental cases per 100,000 inhabitants, in all the studied provinces (Table 2). The greatest correlation coefficient was in Tenerife (r = −0.456) and the lowest was in Zaragoza (r = −0.111). Hospitalization was also inversely correlated to UVI in Zaragoza (r = −0.201), Madrid (r = 0.144), Málaga (r = −0.333), and Tenerife (r = −0.374). There was no association between ICU admission and UVI, except for in Zaragoza (r = −0.174). Inverse correlations between deaths and UVI were statistically significant in Zaragoza (r = −0.219), Málaga (r = −0.264), and Guipuzcoa (r = −0.302).

### 3.3. Correlation between COVID-19 and Temperature

The association between COVID-19 and mean temperature was considered separately, revealing weaker and more heterogenous correlations than were found for UVI (Table 3). Temperature was associated with COVID-19 incidence only in Zaragoza (r = 0.105). Temperature also showed a positive correlation with hospitalization in Guipuzcoa (r = 0.138), but an inverse correlation in Málaga (r = −0.150) and in Tenerife (r = −0.110). ICU admissions showed no correlation with temperature, except for in Zaragoza (r = −0.173). Statistically significant inverse correlations between temperature and deaths caused by COVID-19 were observed in Guipuzcoa (−0.291), Zaragoza (r = −0.166), and Málaga (r = −0.220) (Table 3).

### 3.4. Correlation between COVID-19 and Relative Humidity

Correlations between humidity and COVID-19 variables were very heterogeneous and most were statistically non-significant (Table 4). Inverse correlations were observed between humidity and incidental cases in Guipuzcoa (r = −0.163) and Zaragoza (r = −0.168), while Málaga (r = 0.118) and Tenerife (r = 0.191) showed positive correlations between COVID-19 incidence and humidity. Hospitalization was not associated with humidity, except in Tenerife (r = 0.159). Humidity showed a positive statistically significant correlation with ICU admissions in Guipuzcoa (r = 0.138) and Zaragoza (r = 0.131). Humidity was inversely correlated with mortality only in Guipuzcoa (r = −0.115).

## 4. Discussion

The present study shows that the incidence of COVID-19 was inversely correlated with seasonal variations of UVI in the five provinces studied. Hospitalization and mortality also were inversely correlated with UVI, but not in all the studied provinces. Correlations between temperature and incidental cases were weaker and less consistent, however, a clear negative correlation between temperature and hospitalization was observed in the warmer provinces (Málaga and Tenerife). Deaths were correlated with mean temperatures in Guipuzcoa, Zaragoza, and Málaga. The role played by humidity remains uncertain, because the correlations found with the different COVID-19 variables were inconsistent.

The greatest mean value of daily incidence of COVID-19 cases per 100,000 inhabitants across the study period was found in Madrid (21.39, SD 15.8), followed by Zaragoza (20.98, SD 21.02), Guipuzcoa (17.67, SD 20.35), Málaga (11.53, SD 15.97), and Tenerife (3.82, SD 4.16). Differences were statistically significant between Tenerife and the rest of the provinces. This is in agreement with studies that confirm a lineal correlation between latitude and COVID-19 incidence, without a clear association with temperature or humidity [30]. The higher incidences in Madrid, Zaragoza, and Gipuzkoa might be related to the fact that they are at more northerly latitudes in Spain, which together with higher population density has been found to be an important factor for the spread of COVID-19 [31]. It should be noted that COVID-19 incidence at the beginning of the pandemic in Spain was underestimated due to the low number of PCR tests performed. During the first wave of the disease in Spain, incidental cases dropped after a 4-week lockdown, making it more difficult for the virus to spread. This decrease in the rate of incidence coincided with the middle of spring and a significant rise in solar UV irradiance at the earth’s surface. All the studied provinces began lockdown at the same time, under strict control from the Spanish authorities in an attempt to maintain low COVID-19 incidence levels. A possible explanation for this was that successful control of population movements on the mainland could quickly have an effect on incidence rates; however, higher mean temperatures and increased levels of solar UV radiation reaching the earth’s surface could also contribute to the reduction in cases.

This study’s most relevant and consistent finding is the negative correlation between incidence of COVID-19 cases and the UVI at every studied latitude. These results agree with those published by Páez et al. [32], which were also focused on Spain and found that a daily UV increase was related to a decrease in the accumulated daily growth rate of COVID-19 cases for the two and a half weeks that followed, however, those results were obtained using data from the first three months of the pandemic.

Tang et al. also found a correlation between the average percentage of positive cases of five human coronaviruses (SARS-CoV-2, CoVHKU1, CoVNL63, CoVOC43, and CoV229E) and sunlight UV radiation, but only in the eastern United States and not in the western side [12]. According to data published by the Carlos III National Health Institute, the first wave of infections in Spain during 2020 corresponded to the Sars-CoV-2 SEC8 variant, and in a second wave from the end of summer the variant 20E (EU1) or B.1.177 was predominant until the end of the year. In the third wave from December 2020, the main variant was the Alpha variant. In the case of Spain, the annual distribution of different variants of COVID-19 was also influenced by environmental conditions, in a similar way than observed in previous research [12]. Recently, according to an international analysis, UV exposure had a U-shaped effect on the reproduction number, with a minimum UVI of around 6.3 [33]. The conclusions agree with those of the current study, indicating that UVI during the summer could reduce transmission in temperate regions such as Spain but lead to higher risk in equatorial regions with very high UV exposure, although the previous researchers found this latter result to be unexpected.

In a study performed in Spain during the early months of the pandemic, temperature was described as the main meteorological factor involved in COVID-19 transmission, with Spanish regions that had the highest temperatures showing lower incidences of infection [34]. Another recent study performed in China and the United States also reported that reduction of COVID-19 transmission was strongly correlated with temperature and relative humidity [35]. Another study described how temperatures over 25 °C led to a decrease of the infection rate of 3.7% [95% CI 1.9–5.4] per additional degree [36]. However, our study found no clear correlation between mean temperature and COVID-19 incidence or hospitalization in the studied provinces, showing a weak positive correlation in Zaragoza and a weak negative correlation in Tenerife and Málaga. This could be due to the latter having an annual mean temperature of 23 °C compared to 13 °C in Zaragoza, with lower temperatures potentially promoting indoor activities and social gatherings, leading to virus transmission [22]. Certain other studies have also demonstrated positive correlations between temperature and COVID-19 incidence, such as the research performed by He [37] in nine different Asian cities including in Japan (r = 0.416, *p* < 0.01).

According to our results, it can be stated that the COVID-19 outbreaks that occurred in the spring and at the end of summer support a relationship between UVI, temperature, and incidence of COVID-19. In this context, the end of the spring outbreak was related to UVI over 4 and temperatures over 15 °C, while the summer–autumn outbreak coincided with UVI under 5–6 for all latitudes and temperatures that were again under 20 °C.

Relative humidity may potentially affect COVID-19 transmission. Our results show that in northern provinces, relative humidity correlated negatively with COVID-19 incidence five days later; meanwhile, relative humidity showed a positive correlation in the southern provinces of Málaga and Tenerife. In a recent global analysis of different meteorological factors affecting SARS-CoV-2 transmission, humidity did not show a statistically significant association [33]. However, previous studies support the theory of a negative correlation between relative humidity and COVID-19 transmission [34,35]. An investigation performed in India in April 2020 also considered a 5-day lag between meteorological and COVID-19 variables, and the results indicated that relative humidity and average temperature were negatively associated with COVID-19 [38]. Mozumder et al. concluded that a 1% increase in relative humidity was correlated with a 1.7~3.7% increase in daily confirmed cases of COVID-19 [39], supporting the theory that relative humidity increases COVID-19 transmission. In a study performed in six South Asian countries from March to June 2020, Jain et al. [40] found that high humidity and high temperature increased the transmission of COVID-19 infections, with results applicable to regions with high transmission rates where the minimum temperature is generally over 21 °C. Nicastro et al. [15] clearly explained how the combination of sun radiation, ambient temperature, and humidity can affect the natural lifespan of the virus outdoors. The authors indicated that the effects of both direct and indirect radiation from the sun need to be considered in order to completely explain the effects of UV radiation on life processes. First, there is a UV virucidal effect when the summer period arrives. Then, the UV effect is enhanced in combination with the concomitant process of depletion of water droplets due to solar heat, and/or is reduced in combination with high humidity (because of the larger optical depth). Therefore, summer implies higher temperatures (with heat affecting viruses directly due their thermo sensibility) as well as an environment of lower air humidity(with lower droplet size), which can boost the direct disinfecting power of solar UVB/UVA. In laboratory experiments, under controlled combinations of UV (solar simulator irradiation), temperature, and relative humidity, Shuit et al. [41] found that the most effective treatment to attain virus inactivation was UV radiation. Experimental simulation of three ambient variations similar to those found in summer or winter confirmed the findings of ecological studies revealing that the permanence of the virus had an environmental dependence on these three components. Inactivation of the SARS-CoV-2 virus in relation to temperature and humidity was analyzed under laboratory conditions by Biryukov et al. [42]. At room temperature of 24 °C, the effect of humidity per se was associated with an increase in virus permanence on surfaces at 20% humidity compared with 40% and 60%. This supports other published results [34,35]. However, when the temperature increased from 24 to 35 °C inactivation of the virus increased significantly independently of relative humidity level. [42]. Therefore, it seems that there is a critical temperature–humidity level that affects virus permanence.

Admission to ICU was apparently not related to any of the studied meteorological factors, in agreement with several studies, the severity of the disease being mainly conditioned by individual patient comorbidities [43]. Fang et al. conducted a review that revealed male sex, elderly age, chronic kidney disease, and chronic obstructive pulmonary disease were all significantly associated with severity [44]. Hypertension and diabetes were also found to have been more prevalent among patients who died with the disease.

According to our results, higher mean temperatures and higher levels of ultraviolet radiation correlated with deaths caused by COVID-19. In a retrospective study carried out in Wuhan, temperature was also inversely associated with COVID-19 deaths, whereas diurnal temperature range was positively associated [45].

Environmental data from our work and from other related studies could indicate two distinct effects of seasonal variations on the different COVID-19 outbreaks in 2020 and at the beginning of 2021. First, high UVI and temperature make the air and the earth’s surface comparatively free of the virus on days with UVI over 4–5 and temperatures over 20 °C. Second, spring and summer in Spain, due to their mild temperatures, invite to population to spend more time outside and thus social distancing is increased. According to other studies, the immunomodulatory role of vitamin D, which is clearly influenced by solar UV seasonality, is an evident indirect factor affecting COVID-19 infection rates and severity. From mid-spring to late summer, mean maximal UVI ratings over 8 are to be found in our study locations in Spain, and it is well established that spring and summertime are related to increases in Vitamin D serum levels in humans. Accordingly, autumn and winter, when higher incidences and severity of COVID-19 were manifest, are related to lower outdoor UV levels and lower levels of vitamin D [7,8]. Recent meta-analysis has described the direct role of vitamin D in reducing the severity of COVID-19. Zaazouee et al. [46] showed that vitamin D supplementation in patients with COVID-19 significantly decreased rates of intensive care admissions compared with controls. Similar results were reported by Feiner-Solis et al. [47], and cross-sectional case–control and longitudinal studies have also demonstrated that higher levels of vitamin D were associated with a reduction in COVID-19 risk and severity [48]. Other factors in addition to environmental seasonality can influence the transmission of SARS-CoV-2, including social distancing, protective masks, lockdowns, and vaccination. These are some of the measures that were globally implemented across society and have been fundamental in controlling the COVID-19 pandemic, with several studies showing how implementation of lockdown was crucial to control the transmission of COVID-19 [39]. In addition, numerous individual factors as well as meteorological influences can contribute to the disease’s severity and mortality. Iqbal et al. [49] analyzed the genomic adaptation of SARS-CoV-2 to the UV index variations in 25 countries, reporting that high numbers of distinct recurrent single-nucleotide variants could be attributed only to specific UV index regions. The authors proposed solar UV radiation as one of the driving forces for SARS-CoV-2 differential genomic adaptation [49].

Taking all these limitations into consideration, meteorological factors, especially UV radiation and less probably temperature, seem to have played an important role in incidence levels of COVID-19 in Spain, similar to observations in other countries around the world.

## 5. Conclusions

UV radiation seems to be the most relevant meteorological factor affecting the incidence and hospitalization rates of COVID-19 in Spain throughout the pandemic. Higher UV levels can contribute to lower rates of COVID-19 incidence and hospitalizations. Temperature seems to have a lesser effect that differs in warmer places (inverse correlation) and colder places (direct correlation), while the effect of humidity remains unclear. ICU admissions caused by COVID-19 infection seem not to be influenced by meteorological factors. Higher mean temperatures and higher ultraviolet radiation levels could contribute to reducing numbers of deaths caused by COVID-19.

## Figures and Tables

**Figure 1 ijerph-20-01973-f001:**
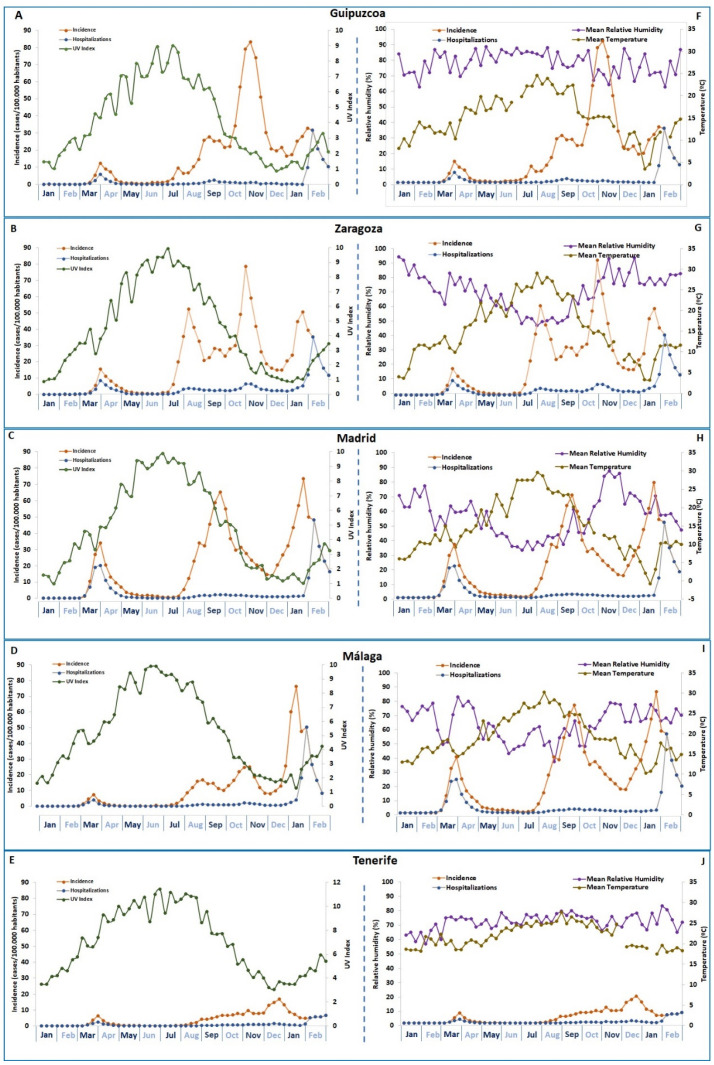
(**A**–**E**) Seasonal variations of mean weekly UVI, COVID-19 incidence rate, and hospitalized cases, with (**F**–**J**) seasonal variations of mean weekly temperature and humidity data for the same locations. COVID-19 incidence rates are also included in (**F**–**J**), to allow better observation of seasonal cycles of meteorological conditions related to COVID-19.

**Table 1 ijerph-20-01973-t001:** Mean UVI, temperature, and humidity per season in the five studied provinces, with the incidence of COVID-19 variables from winter 2020 to the end of winter 2021. Global mean data followed by SD are shown for each variable and localization.

		Guipuzcoa	Zaragoza	Madrid	Málaga	Tenerife
		Mean UV Index				
2020	Winter	2.6	2.5	2.9	3.5	5.4
	Spring	6.8	7.6	7.9	8.0	9.9
	Summer	6.5	7.3	7.7	7.9	9.6
	Autumn	1.8	2.0	2.4	2.9	4.6
2021	Winter	2.0	2.0	2.2	2.8	4.5
	Mean (SD)	4.0 (2.5)	4.3 (2.9)	4.6 (2.9)	5.0 (2.7)	5.0 (2.7)
		Mean temperature (°C)				
2020	Winter	11.5	9.7	9.8	15.4	19.7
	Spring	17.6	19.3	18.4	20.8	21.9
	Summer	21.7	24.9	24.9	26.7	25.4
	Autumn	13.3	12.0	11.1	17.5	22.3
2021	Winter	10.4	9.0	6.9	14.0	18.6
	Mean (SD)	14.9 (4.7)	15.0 (6.9)	14.2 (7.3)	18.9 (5.1)	21.6 (2.6)
		Mean Relative Humidity (%)				
2020	Winter	76.9	79.4	63.1	69.2	67.1
	Spring	81.9	67.3	50.4	59.9	72.1
	Summer	81.8	52.7	41.9	54.4	76.4
	Autumn	74.8	78.8	70.5	68.4	72.9
2021	Winter	74.9	78.8	57.7	70.4	73.9
	Mean (SD)	78.4 (3.6)	71.4 (11.6)	56.4 (11.1)	64.5 (7.0)	72.5 (3.4)
		COVID-19 Incidence (number of cases/100,000 habitants)				
2020	Winter	1.59	1.89	6.05	1.14	0.91
	Spring	2.20	2.96	5.51	0.74	0.69
	Summer	14.86	25.91	27.91	8.37	2.57
	Autumn	42.44	33.09	24.61	15.07	9.97
2021	Winter	22.57	30.60	42.89	38.91	6.12
	Mean (SD)	16.73 (16.9)	18.89 (15.3)	21.39 (15.8)	12.84 (15.7)	4.05 (4.0)
		COVID-19 Hospitalizations (number cases/100,000 habitants)				
2020	Winter	0.75	1.05	4.07	0.67	0.42
	Spring	0.52	1.18	2.11	0.24	0.26
	Summer	0.80	2.24	1.15	0.54	0.24
	Autumn	0.63	3.32	1.33	1.13	0.92
2021	Winter	10.92	13.75	17.00	15.92	3.25
	Mean (SD)	2.73 (4.6)	4.31 (5.4)	5.13 (6.7)	3.70 (6.8)	1.02 (1.3)
		COVID-19 Intensive Care Units patients (number cases/100,000 habitants)				
2020	Winter	0.07	0.16	0.33	0.08	0.08
	Spring	0.02	0.13	0.17	0.02	0.03
	Summer	0.09	0.11	0.06	0.05	0.05
	Autumn	0.08	0.29	0.07	0.08	0.14
2021	Winter	0.01	0.32	0.16	0.27	0.12
	Mean (SD)	0.05 (0.04)	0.20 (0.10)	0.16 (0.11)	0.10 (0.10)	0.08 (0.05)
		COVID-19 Deaths (number cases/100,000 habitants)				
2020	Winter	0.08	0.15	0.55	0.07	0.04
	Spring	0.38	0.65	0.81	0.11	0.08
	Summer	0.12	0.44	0.18	0.07	0.02
	Autumn	0.70	0.94	0.33	0.25	0.14
2021	Winter	0.39	0.65	0.44	0.59	0.14
	Mean (SD)	0.39 (0.29)	0.56 (0.33)	0.46 (0.27)	0.22 (0.09)	0.08 (0.05)

**Table 2 ijerph-20-01973-t002:** Correlations between UVI and COVID-19 incidence rate, hospitalized cases, ICU-admitted cases, and deaths per 100,000 inhabitants.

	Cases	Hospitalized	ICU-Admitted	Deaths
	/100,000 inh.	/100,000 inh.	/100,000 inh.	/100,000 inh.
GUIPUZCOA	−0.242 *p* < 0.001	N.S.	N.S.	−0.302 *p* < 0.001
ZARAGOZA	−0.111 *p* = 0.02	−0.201 *p* < 0.001	−0.174 *p* < 0.001	−0.219 *p* < 0.001
MADRID	−0.188 *p* < 0.001	−0.144 *p* = 0.005	N.S.	N.S.
MÁLAGA	−0.362 *p* < 0.001	−0.333 *p* < 0.001	N.S.	−0.264 *p* < 0.001
TENERIFE	−0.456 *p* < 0.001	−0.374 *p* < 0.001	N.S.	N.S.

**Table 3 ijerph-20-01973-t003:** Correlations between mean temperature and COVID-19 incidence rate, hospitalized cases, ICU-admitted cases, and deaths per 100,000 inhabitants.

	Cases /100,000 inh.	Hospitalized /100,000 inh.	ICU-Admitted /100,000 inh.	Deaths /100,000 inh.
GIPUZCOA	N.S.	0.138 *p* = 0.007	N.S.	−0.291 *p* < 0.001
ZARAGOZA	0.105 *p* = 0.031	N.S.	−0.173 *p* < 0.001	−0.116 *p* = 0.019
MADRID	N.S.	N.S.	N.S.	N.S.
MÁLAGA	N.S.	−0.150 *p* = 0.003	N.S.	−0.220 *p* < 0.001
TENERIFE	N.S.	−0.110 *p* = 0.031	N.S.	N.S.

**Table 4 ijerph-20-01973-t004:** Correlations between mean humidity and COVID-19 incidence rate, hospitalized cases, ICU-admitted cases, and deaths per 100,000 inhabitants.

	Cases /100,000 inh.	Hospitalized /100,000 inh.	ICU-Admitted /100,000 inh.	Deaths /100,000 inh.
GIPUZCOA	−0.163 *p* = 0.001	N.S.	0.138 *p* = 0.005	−0.115 *p* = 0.02
ZARAGOZA	−0.168 *p* = 0.01	N.S.	0.131 *p* = 0.008	N.S.
MADRID	NS	N.S.	N.S.	N.S.
MÁLAGA	0.118 *p* = 0.015	N.S.	N.S.	N.S.
TENERIFE	0.191 *p* < 0.001	−0.159 *p* = 0.002	N.S.	N.S.

## Data Availability

The datasets used and/or analysed during the current study are available from the corresponding author on reasonable request.

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
