# Peer review of "Correlation between UV Index, Temperature and Humidity with Respect to Incidence and Severity of COVID 19 in Spain"

_ijerph, 2023, doi:10.3390/ijerph20031973_

Round 1

Reviewer 1 Report (Previous Reviewer 3)

All issues raised by the reviewers have been addressed, thus I have no more comments.

Author Response

Dear Reviewer, 

thanks a lot for your review and we are happy that all corrections you indicated to us are ok for you. 

Authors

Reviewer 2 Report (New Reviewer)

Its interesting study, but major concern is there with the manuscript as follow:

- There is no discussion for UV A.

- The authors mentioned that "UVB radiation is crucial 56 for vitamin D synthesis in the skin" and also "several studies support the association 58 between low serum vitamin D levels and more severe COVID-19 disease" this statement is in contrast with the current result. Please discuss more about it and the effect of vitamin D on COVID-19 protection.

- In this period of time what were the variants of COVID-19? It seems that 2 different varients were spread in this period.

- Please insert or combine the incidence graph with the temperature and humidity graph in fig. 1.

- I personally can not realize the mechanism of correlation between humidity and severity. Please explain more.

- Is there any in vitro experiment on the relation between UVI, temperature, and humidity with COVID-19 severity?

Author Response

Answer letter to reviewer 2

Its interesting study, but major concern is there with the manuscript as follow:

Referee: - There is no discussion for UV A.

Authors: Referee is right in taking into account of UVA part of solar spectrum and its possible role together to UVB photons in inactivation of RNA of DNA viruses. There is not many publications about the effect of UVA or UVB independently to the virus inactivation. At priori, most of literature is related to the solar UV level correlations to virus infections and severity in an ecological point of view (MS ref 12,13, 14,). There are some publications related to UVB effect on virus inactivation (ref 10 and 22) as we indicate in introduction section, and the UV index has been adapted to most meteorological agencies as UV indicator of biological damaging effect of UV radiation. It is clear that UV index is based on erythemal effect to human skin, but due that erythema has a similar action spectrum at earth surface level than the action spectrum for DNA damage this is the main reason for adopting UVI as indicator.

Regarding to UVA, Nicastro et al.  and Biasin et al. have published the action spectrum of COVID19 inactivation and wavelengths of 366 and 405 nm are effective for that inactivation. Au-thors indicated that SARS-COV-2 as well as other RNA viruses are particularly sensitive to solar UV radiation. UVA necessary doses for virus inactivation were 104 higher than UVC although authors data confirm that summer UV exposure could have enough UVB and UVA dose for virus inactivation in less than 2 minutes outdoor..

Nicastro F, Sironi G, Antonello E, Bianco A, Biasin M, Brucato JR, Ermolli I, Pareschi G, Salvati M, Tozzi P, Trabattoni D, Clerici M. Solar UV-B/A radiation is highly effective in inactivating SARS-CoV-2. Sci Rep. 2021 Jul 20;11(1):14805. doi: 10.1038/s41598-021-94417-9.

Biasin M, Strizzi S, Bianco A, Macchi A, Utyro O, Pareschi G, Loffreda A, Cavalleri A, Lualdi M, Trabattoni D, Tacchetti C, Mazza D, Clerici M. UV and violet light can Neutralize SARS-CoV-2 Infectivity. J Photochem Photobiol. 2022 Jun;10:100107. doi: 10.1016/j.jpap.2021.100107. Epub 2022 Jan 8.

Referee:The authors mentioned that "UVB radiation is crucial 56 for vitamin D synthesis in the skin" and also "several studies support the association 58 between low serum vitamin D levels and more severe COVID-19 disease" this statement is in contrast with the current result. Please discuss more about it and the effect of vitamin D on COVID-19 protection.

Authors: In that case we can affirm, from results derived of other works that summer spring and summer time implied an increment of total UV radiation incoming to our study localizations in Spain and it is well stablished that spring and summer time are related to an increase in Vitamin D serum levels on humans. So, autumn and winter time, where higher incidence an severity of COVID was reached is related to lower outdoor UV levels and lower levels of vitamin D. References 7-8 of the MS confirmed this thesis and more recent papers describe the role of vitamin D directly in prevention of COVID 19 severity. Zaazouee et al (2022) in a systematic review and meta-analysis showed that vit D supplementation in patients with COVID 19 decreased significantly the intensive care admission compared to control. Similar results have been shown by Feiner-Solis et al (2022) and in cross sectional case-control and longitudinal studies also demonstrated a reduction in COVID risk and severity (Martineau 2022).

This paragraph has been included in the discussion section

-Zaazouee MS, Eleisawy M, Abdalalaziz AM, Elhady MM, Ali OA, Abdelbari TM, Hasan SM, Almadhoon HW, Ahmed AY, Fassad AS, Elgendy R, Abdel-Baset EA, Elsayed HA, Elsnhory AB, Abdraboh AB, Faragalla HM, Elshanbary AA, Kensara OA, Abdel-Daim MM. Hospital and laboratory outcomes of patients with COVID-19 who received vitamin D supplementation: a systematic review and meta-analysis of randomized controlled trials. Naunyn Schmiedebergs Arch Pharmacol. 2022 Dec 12:1–14. doi: 10.1007/s00210-022-02360-x. Epub ahead of print.

-Feiner Solís Á, Avedillo Salas A, Luesma Bartolomé MJ, Santander Ballestín S. The Effects of Vitamin D Supplementation in COVID-19 Patients: A Systematic Review. Int J Mol Sci. 2022 Oct 17;23(20):12424. doi: 10.3390/ijms232012424. 

-Martineau AR. Vitamin D in the prevention or treatment of COVID-19. Proc Nutr Soc. 2022 Nov 11:1-8. doi: 10.1017/S0029665122002798. Epub ahead of print.

Referee:- In this period of time what were the variants of COVID-19? It seems that 2 different varients were spread in this period.

Authors: Referee is right: From data of National Health Institute Carlos III, in Spain, during 2020, the first wave corresponded to Sars-Cov-2 SEC8 variant and in a second wave by end of summer, the variant 20E (EU1) or B.1.177 was predominant until the end of the year in which by December of 2020 in the third wave, the variant corresponded to Alpha variant.  

This has been included in the new version of the MS

Referee:- Please insert or combine the incidence graph with the temperature and humidity graph in fig. 1.

Authors: We have combined, as well as that UVI graph, the incidence with the temperature and humidity in order to better observation of annual changes in incidence with respect to the ambient variables.

Referee:- I personally can not realize the mechanism of correlation between humidity and severity. Please explain more.

Authors; In the paper of Nicastro et al (2021) is clearly explained how the combination of sun radiation, ambient temperature as well as humidity can affect the natural permanence of virus outdoor. Authors indicate that the effects of both direct and indirect radiation from the Sun needs to be considered in order to completely explain the effects of UV radiations in life processes. First, the UV virucidal effect when summer period arrives and the UV effect is enhanced in combination with the concomitant process of water droplets depletion because of solar heat and/or reduced in combination with high-humidity because of the larger optical depth). So, summer implies higher temperature (heat affecting viruses directly due their thermosensibility) as well as that lower air humidity environment (with lower droplet size) can boost the direct disinfecting power of solar UVB/UVA.

However in laboratory controlled conditions, the permanence of SARS-CoV-2 in different surfaces depended on combination temperature and relative humidity. At 24º, an increase of relative humidity from 20 up to 80% decreased permanence of virus in the surface. So, higher inactivation of virus was found at higher RH. However, when temperature was increased to 35º the inactivation rate of virus is similar at 20 and 40% of relative humidity. So, droplets sizes and temperature transmittance in high relative humidity may be responsible of changes in the inactivation rates.

This has bieen included in the discussion section.

Referee:- Is there any in vitro experiment on the relation between UVI, temperature, and humidity with COVID-19 severity?

Authors: Shuit et al reported laboratory measurements regarding de combination of UV (solar simulator experiments) and temperature and relative humidity control. In that experiments authors find the most relevant effect for virus inactivation was UV radiation and simulation of temperature and RH in summer and winter time confirm ecological works with and environmental dependence for virus permanence depending on these three components.

Recently the inactivation of SARS-CoV-2 virus related to temperature and humidity has been analyzed in laboratory conditions by Biryukov et al (2020) and other works there in. The effect of humidity per se implies an increase in virus permanence in surfaces at 20% humidity compared to 40% and 60%. However, when temperature is increased the virus inactivation is decreased significantly independent of RH of 20-40%. At higher RH than 60% the higher temperature increases the virus inactivation than lower humidity.

-Schuit M, Ratnesar-Shumate S, Yolitz J, Williams G, Weaver W, Green B, Miller D, Krause M, Beck K, Wood S, Holland B, Bohannon J, Freeburger D, Hooper I, Biryukov J, Altamura LA, Wahl V, Hevey M, Dabisch P. Airborne SARS-CoV-2 Is Rapidly Inactivated by Simulated Sunlight. J Infect Dis. 2020 Jul 23;222(4):564-571. doi: 10.1093/infdis/jiaa334. 

-Biryukov J, Boydston JA, Dunning RA, Yeager JJ, Wood S, Reese AL, Ferris A, Miller D, Weaver W, Zeitouni NE, Phillips A, Freeburger D, Hooper I, Ratnesar-Shumate S, Yolitz J, Krause M, Williams G, Dawson DG, Herzog A, Dabisch P, Wahl V, Hevey MC, Altamura LA. Increasing Temperature and Relative Humidity Accelerates Inactivation of SARS-CoV-2 on Surfaces. mSphere. 2020 Jul 1;5(4):e00441-20. doi: 10.1128/mSphere.00441-20.

We would like to thank to referee for this constructive review that increases quality of the manuscript.

Round 2

Reviewer 2 Report (New Reviewer)

The revised version is acceptable now.

This manuscript is a resubmission of an earlier submission. The following is a list of the peer review reports and author responses from that submission.

Round 1

Reviewer 1 Report

The authors preset their study on the correlation between UV index, humidity and temperature with COVID incidence and hospitalization in Spain.

The methods are well described, but some of the correlations they found are not new (or only new to the studied provinces in Spain, but not in general). Also, the provinces they chose for the study are not easily comparable, because the population of the provinces are different and one of them (Tenerife) is an island and for this reason the COVID spread is diferent. The authors compensated the difference in population with dividing the number of cases among the total population, but this is a factor, which influences the spread of COVID, because in cities an provinces with high population, this highly contagious disease can spread easier and faster. In contrast, in an island, such as Tenerife, it is more protected, because the disease from the peninsula cannot just reach the island that easy. This effect is reflected in the results of the authors, as the correlations they usually found with other provinces were different in case of Tenerife.

On page 4, the authors describe how long delays are applied with the analysis of correlation between the meteorological circumstances and the COVID incidence, hospitalization etc., but it is not clear, how they chose these timepoints (5 vs 9 vs 15 days).

As the authors describe in the Discussion of the manuscript, their findings are not new, the negative correlation between the COVID-19 incidence was already described, as well as the influence of the latitude. In addition, the not clear association with temperature and humidity was also published.

For the above mentioned reasons I do not suggest this publication for publishing in International Journal of Environmental Research and Public Health.

Author Response

Dear Reviewer,

Thanks a lot for your comments. As you indicated, some previous works treat similar topics and reach similar conclusions. In our case, data from Spain situation are very important because in comparison with other European countries the latitude situation in the south of Europe and the big differences in latitudes inside our country (Tenerife Islands vs. peninsula) is a very good example for comparation of COVID incidences in relation with environmental variables. Is important to point out this kind of works because it is of absolute importance in terms of decisions in population control strategies in this kind of pandemic situations. Higher UV and temperature associated seasons are important to help to control situations due to an outdoor environment more convenient to open restrictions of people mobility. Moreover, our study is able to point out with a very good precision year dates for make this kind population decisions, in which, when maximal UV index is over 5 in spring, restrictions can be opened as results indicated and the contrary at the end of the summer.

Regarding your comment On page 4, “the authors describe how long delays are applied with the analysis of correlation between the meteorological circumstances and the COVID incidence, hospitalization etc., but it is not clear, how they chose these timepoints (5 vs 9 vs 15 days)”.

We have considered this correlation date point due to the normal patter of the history of the Covid 19 illness during the pandemic situation in the year 2020. Following references 26 and 27 included in the text, we have considered that human-to-human contagion delays 4-7 days and the outdoor environment data has to be considered with this period of advance. Mean time to hospitalization and ICU admission and finally dead takes a delay of days considered for the correlations to environment variables.

  • M Linton NM, Kobayashi T,  Yang Y,  Hayashi K,  Akhmetzhanov AR, Jung SM, et al. Incubation Period and Other Epidemiological Characteristics of 2019 Novel Coronavirus Infections with Right Truncation: A Statistical Analysis of Publicly Available Case Data. J Clin Med. 2020;9:538. https://doi.org/10. 3390/jcm9020538.
  • Yang X, Yu Y, Xu J, Shu H, Xia J, Liu H et al. Clinical course and outcomes of critically ill patients with SARS-CoV-2 pneumonia in Wuhan, China: a single-centered, retrospective, observational study. Lancet Respir Med. 2020;8:475-481 https://doi.org/10.1016/S2213-2600(20)30079-5.

Again thanks a lot for your comments

Reviewer 2 Report

Correlation between UV index, temperature and humidity with respect to incidence and severity of COVID 19 in Spain

In this study the authors have analysed the correlation between the environmental factors (ultraviolet index (UVI), temperature and humidity) and the incidence, severity and mortality of COVID 19 in different latitudes in Spain. The authors have collected data from January 2020 to February 2021 covering all the latitudes and seasons.  The results of the study show that higher UVI can lead to lower incidence of COVID 19 and hospitalization.  Though the study area is limited to Spain covering latitude from 28 to 43 the findings of this study will be useful for researchers working on the epidemiology of COVID 19 and other respiratory viral infections.  The article is suitable for publication. Corrections suggested below may be carried out before final submission.

Corrections/modifications required

Section

Line No.

Corrections

Abstract

21

used to analyzed   to be corrected as  used to analyze   

Introduction

31

pandemic is believed to start    to be corrected as  pandemic is believed to have started   

32

Mild cases, commonly ........... throat pain. - Not clear. Please check the sentence.

36

Diagnosis is ...........phases.  - Not clear. Please check the sentence.

40

Nowadays.......... another. - Not clear. Please check the sentence.

42

disease – causing virus  to be corrected as  disease – causing viruses

infected people   to be corrected as  infected person  

45

Sun radiation ........... radiation. - Not clear. Please check the sentence.

58

effective  decontaminating air  to be corrected as  effective in  decontaminating air 

59

does not content UVC  to be corrected as  does not contain UVC 

64

Evidence .......... increases. - - Not clear. Please check the sentence.

Materials and methods

84

Data were obtained   to be corrected as  Data were obtained   from

Discussion

181

According to ........... provinces (Málaga and Tenerife). – Rephrase the sentence.

188

The greatest media   to be corrected as  The greatest mean  

199

This studio’s   to be corrected as  This study’s  

204

the three first months of   to be corrected as  the first three months of  

218

Recently, a weak moderate......... additional degree. -  Not clear. Please check the sentence.

223

wean negative   to be corrected as   weak  negative  

   Note: Corrected/added words are underlined.

Author Response

Authors Response to Reviewer 2

Dear Reviewer,

Thanks a lot for your comments.

Corrections/modifications required

Section

Line No.

Corrections

Abstract

21

used to analyzed   to be corrected as  used to analyze   

Authors: Corrected

Introduction

31

pandemic is believed to start    to be corrected as  pandemic is believed to have started   

Authors: Corrected

32

Mild cases, commonly ........... throat pain. - Not clear. Please check the sentence.

Authors:Corrected, This is the new sentence included in the new version of the MS: “In the case of middle-aged patients, including adolescents, young people and adults, illness symptoms such as cough, fever and sore throat are common

36

Diagnosis is ...........phases.  - Not clear. Please check the sentence.

Authors:Corrected, This is the new sentence included in the new version of the MS

“The diagnosis is first made on the basis of the symptoms presented by the patient, and must then be confirmed by PCR test on nasopharyngeal sampling or by serological test.”

40

Nowadays.......... another. - Not clear. Please check the sentence.

Authors:Corrected, This is the new sentence included in the new version of the MS

The principal mode for virus spread carrying infectious virus is through exposure to respiratory fluids carrying infectious virus. Exposure occurs in three principal ways: (1) inhalation of very fine respiratory droplets and aerosol particles, (2) deposition of respiratory droplets and particles on exposed mucous membranes in the mouth, nose, or eye by direct splashes and sprays, and (3) touching mucous membranes with hands that have been soiled either directly by virus-containing respiratory fluids or indirectly by touching surfaces with virus on them

42

disease – causing virus  to be corrected as  disease – causing viruses infected people   to be corrected as  infected person  

Authors: corrected

45

Sun radiation ........... radiation. - Not clear. Please check the sentence.

Authors:Corrected, This is the new sentence included in the new version of the MS

Approximately 99% of the solar radiation reaching the Earth's surface is contained in the region between 0.2 and 3.0 µm, from UV to Infrared radiation. UV radiation levels oscillate along the different seasons, with greater levels during spring or summer, and lower along autumn and winter. Clouds or pollution can contribute in changes of UV radiation reaching the earth surface [5].

58

effective  decontaminating air  to be corrected as  effective in  decontaminating air 

Authors: Corrected

59

does not content UVC  to be corrected as  does not contain UVC 

Authors: corrected

64

Evidence .......... increases. - - Not clear. Please check the sentence.

Authors:Corrected, This is the new sentence included in the new version of the MS

It has been also found literature evidence that COVID-19 spreading outdoor can be partially suppressed as temperature and humidity increases

Materials and methods

84

Data were obtained   to be corrected as  Data were obtained   from

Authors:Corrected

Discussion

181

According to ........... provinces (Málaga and Tenerife). – Rephrase the sentence.

Authors:Corrected, This is the new sentence included in the new version of the MS

In case of correlations between incidental cases with respect to temperature, correlations were less consistent and weaker in terms of incidental cases, but clearly it was observed an inverse correlation between temperature and hospitalizations in the less cold provinces (Málaga and Tenerife).

188

The greatest media   to be corrected as  The greatest mean  

Authors:Corrected,

199

This studio’s   to be corrected as  This study’s  

Authors:Corrected,

204

the three first months of   to be corrected as  the first three months of  Authors:Corrected,

218

Recently, a weak moderate......... additional degree. -  Not clear. Please check the sentence.

Authors:Corrected, This is the new sentence included in the new version of the MS

Recently, a moderate negative relationship was calculated between the estimated infection rate and temperatures warmer than 25ºC, that leads to a decrease of 3.7% of the infection rate [95% CI 1.9-5.4] per additional degree.

223

wean negative   to be corrected as   weak  negative  

Authors:Corrected,

Again thanks a lot for your corrections. All changes have been included in the new versión of the MS.

Reviewer 3 Report

The manuscript belongs to the category of research articles focusing on proving the relation of COVID-19 incidence and severity with UV Index, temperature and humidity in different locations in Spain. The authors found a statistically significant inverse correlation of UVI with incidence rates for all sites and with hospitalizations for all sites but one. For other meteorological factors, results are not consistent and are discussed. The manuscript fits well with the journal's scope. However, prior publishing several problems should be clarified and discussion should be improved. Also, there are many “typos” in the manuscript. Thus, the reviewer recommends a major revision of the article. All the limitations should be discussed and clearly stated in the manuscript.
The first problem is the underestimation of incidence cases, as the number of PCR tests performed is lower than the real cases, which surely affects the results. Even with the wide testing policy, many of the cases can be mistaken with another disease by the doctors and were not taken into consideration for testing.
The second problem is that probably in Spain, especially at the beginning of the COVID-19 pandemic, as in many other countries, lockdowns were performed, thus assessing the spreading paths of the virus can be difficult and probably bared with some errors. This should be a part of the discussion.
Also, there is a discussion of the statistical significance of correlation coefficients according to UV Index, but the reviewer found no discussion about how strong this correlation is. The correlation coefficient of less than 0.5 seems to be not strong enough to formulate the evidence.
Furthermore, in 2020 and the beginning of 2021 the dominant variant of COVID-19 was variant “alpha”, thus the results refer only to that variant and cannot be easily transformed to all mutations – it should be mentioned in the manuscript.
Detailed comments:
Line 44: What about lockdowns and “stay at home” policy? Where there performed in the considered regions?
Line 49 – 52: UV Index is not a measure of the intensity of UV radiation reaching the earth's surface. It is the intensity of UV radiation weighted with the erythemal action spectrum, thus it is the measure of the intensity of UV radiation reaching the earth's surface, which causes erythema (mostly UVB).
Line 59: UV radiation from the sun does contain UVC, but it is absorbed by the atmosphere and does not reach the earth's surface. Also, what is the role of UVA and UVB in the inactivation of SARS-CoV-2? The references are missing.
Line 104: the word “climate” refers to long-period changes in meteorological variables, thus cannot be used here. Probably the term “weather” would be better. Also in the other parts of the text “climatic factor” should be replaced with “meteorological factor”.

Author Response

Authors Response to Reviewer 3

Dear Reviewer,

First of all we would like to thank sincerely to you for the deep correction and proposals that will improve the quality of our work.  I will answer point by point your corrections and recommendations.

Corrections/modifications required

The manuscript belongs to the category of research articles focusing on proving the relation of COVID-19 incidence and severity with UV Index, temperature and humidity in different locations in Spain. The authors found a statistically significant inverse correlation of UVI with incidence rates for all sites and with hospitalizations for all sites but one. For other meteorological factors, results are not consistent and are discussed. The manuscript fits well with the journal's scope. However, prior publishing several problems should be clarified and discussion should be improved. Also, there are many “typos” in the manuscript. Thus, the reviewer recommends a major revision of the article. All the limitations should be discussed and clearly stated in the manuscript.

The first problem is the underestimation of incidence cases, as the number of PCR tests performed is lower than the real cases, which surely affects the results. Even with the wide testing policy, many of the cases can be mistaken with another disease by the doctors and were not taken into consideration for testing.

Authors: The incidence cases, as well as the hospitalization and, ICU and patient death data were obtained from the National Epidemiology Centre (CNE) depending the Official Carlos III Institute of Health in Spain. This is the most relevant institution in Spain, depending of the General Health Ministery. All incidence cases included in the data bases were confirmed by PCR test of blood test. We agree with the reviewer that specially at the beginning of the pandemic situation, after the middle of March, much more incidence rates must be observed but not all of them were confirmed by PCR and the incidence rate was lower. That is a limitation. However the hospitalization rate as well as the ICU and deaths incidences are 100% verified by the ministery after reporting of the different health centers in Spain. We are sure that the incidence rate at March was as higher than the incidence rates at the end of August where much more test were performed in population. But what is clear is that the this firs pandemic situation “namely first wave of covid-19” showed a decrease in rate correlated with the increment of the solar UV radiation irradiance at earth surface by the end of April. And again a negative correlation with  the decrease of UV radiation irradiance by the end of August and the start of the second COVID-19 wave.  

The second problem is that probably in Spain, especially at the beginning of the COVID-19 pandemic, as in many other countries, lockdowns were performed, thus assessing the spreading paths of the virus can be difficult and probably bared with some errors. This should be a part of the discussion.

Authors: We have included this paragraph in the new version of the MS.

Another situation to take into account, specially at the beginning of the pandemic situation is that the relative strong decrease in the incidence rate, that coincided in Spain with the middle spring season and with a significant increase in the solar UV irradiance at the earth surface, was the strong population control that happened with the 4 weeks lockdown and therefore, assessing the spreading path of the virus more difficult and the incidence rate was significantly decreased.

Also, there is a discussion of the statistical significance of correlation coefficients according to UV Index, but the reviewer found no discussion about how strong this correlation is. The correlation coefficient of less than 0.5 seems to be not strong enough to formulate the evidence.

Authors: We have followed the interpretation levels of correlation coefficients according to the MS of

P Schober, C Boer, L A Schwartencluded Correlation Coefficients: Appropriate Use and Interpretation Anesth Analg.  2018 May;126(5):1763-1768. . doi: 10.1213/ANE.0000000000002864.

The show the limits for a good interpretation and of course, weak correlations (0.1-0.39 in the CC) have to be taken with care in order to formulate evidences but our CC were of statistical significance. May be with higher data set and latitude sets the weak correlation could pass to moderate or strong. Our data set selected provinces of different parts of Spain with regarding to very high number of population and the significance was observed as shown in the tables. We have included this reference in the new version of the MS.

Schober P, Boer C, Schwartencluded L A. Correlation Coefficients: Appropriate Use and Interpretation Anesth Analg. 2018;126:1763-1768. https://doi.org/10.1213/ANE.0000000000002864.

Furthermore, in 2020 and the beginning of 2021 the dominant variant of COVID-19 was variant “alpha”, thus the results refer only to that variant and cannot be easily transformed to all mutations – it should be mentioned in the manuscript.

Authors: Thanks a lot for the comment. A new sentence has been included in the new version of the MS.

Finally, there is another new important fact that could be a limitation for evidences reached in our work. Almost no new research has been performed in how can affect the annual environmental variations the infection rates of the new variants of COVID-19 from variant alpha at the beginning of the pandemic. Iqbal et al. 2022 has recently analyzed the differential genomic adaptation of the SARS-CoV-2 to the variations of UV index. Authors analyzed 2500 full-grade genomes from 5 different UVI regions (25 countries) and they found that the highest number of recurrent single nucleotide variants that were distinct were uniquely attributed to the specific UVindex regions, proposing that solar UV radiation as one of the driving forces for SARS-Cov2 differential genomic adaptation (42). Taking all limitations into consideration, meteorological factors, especially UV radiation and probably temperature, seem to play a role in the COVID-19 in Spain as observed for other countries in the world.

We have included this reference in the new version of the MS.

Iqbal N, Rafiq M, Masooma, Tareen S, Ahmad M, Nawaz F, et al. The SARS-CoV-2 differential genomic adaptation in response to varying UVindex reveals potential genomic resources for better COVID-19 diagnosis and prevention. Front Microbiol. 2022;13:922393. https://doi.org/10.3389/fmicb.2022.922393.

Detailed comments:
Line 44: What about lockdowns and “stay at home” policy? Where there performed in the considered regions?

Authors: In Spain the lock down was absolutely strict since 20th of march 2020. A very strong social control was made from Spanish authorities and the population must follow, under a very strong control, this lock down at home. This happened for all latitudes in the MS. However, only in Canary Islands, where Tererife data has been included, the free movement of people, with some exceptions was early made 2 weeks before than the rest of localizations of the Spanish peninsula. This occurred due to the significant lower incidence rates in Canary Islands as observed in the MS. One of the initial hypothesis was that in case of Islands, better control of population movement from outside has rapid effect on incidence rates and second the latitude with higher mean temperature and solar UV reaching the earth surface.

This has been included in discussion of the new version of the MS

Line 49 – 52: UV Index is not a measure of the intensity of UV radiation reaching the earth's surface. It is the intensity of UV radiation weighted with the erythemal action spectrum, thus it is the measure of the intensity of UV radiation reaching the earth's surface, which causes erythema (mostly UVB). Line 59: UV radiation from the sun does contain UVC, but it is absorbed by the atmosphere and does not reach the earth's surface. Also, what is the role of UVA and UVB in the inactivation of SARS-CoV-2? The references are missing.

Authors: Referee is right in case that UVindex is an biological endpoint of the UV radiation reaching the earth in terms of erythema. We have included this new paragraph in which we include explanation based on different classic and recent publications about the solar UV effect on SARS Cov-2 inactivation outdoor and the use of UVI index for correlations between solar UV radiation and COVID 19 infections patterns and virus inactivation. We have included new references in the MS to support the evidences.

In regard to the environmental effect of UV radiation, UVC has been shown to be very effective in decontaminating the air from SARS-CoV-2 [8]. Although UV radiation from the sun does not contain UVC, solar UV radiation has been proposed to be an envi-ronmental determinant in shaping the transmission of COVID-19 at the seasonal time scale [9] and its has been clearly negatively correlated with infection levels  and mor-tality of patients [10-13]

Ultraviolet Index (UVI) measures the intensity of UV radiation with skin erythemal potential that is found at a daytime and place on the earth surface at ground level [14]. Since most of the UV part of the solar spectrum that promotes skin erythema is UVB radiation, and it is correlated to DNA damage, it is actually considered the main variable for measurements of damaging solar potential in terms of biological effects on humans and it use has been standardize by almost all the meteorological agencies in the world [15]. The Ultraviolet Index (UVI) measures the intensity of UV radiation with skin erythemal potential that is found at a daytime and place on the earth surface at ground level [16]. The solar erythemal irradiance is obtained by the integration in the region between 290-400 nm of the UV radiation at ground level, weighted by the erythemal action spectrum. The DNA-damage UV index is an integration between 256-370nm at ground level weighted with the DNA-damage action spectrum [17,18]. So, different authors related seasonality of infection patterns as well as the inactivation of the SARS-CoV-2 virus as a function of the standard UVI [19,20].

Here I show the similarities of DNA and Erythemal action spectrum and the main UVB dependence of both action spectra.

Line 104: the word “climate” refers to long-period changes in meteorological variables, thus cannot be used here. Probably the term “weather” would be better. Also in the other parts of the text “climatic factor” should be replaced with “meteorological factor”.

Authors: We agree with reviewer. Thanks a lot for the correction.

Again thanks a lot for your comments and discussion.

Round 2

Reviewer 3 Report

The authors improved the manuscript significantly, but after adding a new text,  some English corrections are required. Also, not all of the reviewer's comments have been included in the present version of the manuscript. Thus I recommend to accept article after minor revision.

Detailed comment:

Line 55: authors still claim that "Although UV radiation from the sun does not contain UVC (...)", which is not true. UV radiation from the sun contains UVC, but it is blocked by the atmosphere (namely ozone layer) and does not reach the earth's surface.

Author Response

Reviewer 3 Answer letter

Comments and Suggestions for Authors

The authors improved the manuscript significantly, but after adding a new text,  some English corrections are required. Also, not all of the reviewer's comments have been included in the present version of the manuscript. Thus I recommend to accept article after minor revision.

Authors: Thanks a lot by your new review of the MS and sorry very much for the mistake in editing the corrected version of the MS. You are right and corrected paragraph following indications of the Reviewer 2 were not included. This has been included in this new version of the MS and the corrections of the added new text (in yellow) has been revised by native speaker.

Detailed comment:

Line 55: authors still claim that "Although UV radiation from the sun does not contain UVC (...)", which is not true. UV radiation from the sun contains UVC, but it is blocked by the atmosphere (namely ozone layer) and does not reach the earth's surface.

Authors: Again you are right, the UV reaching to the earth contains UVC but is blocked by the ozone layer. This has been corrected in the new version of the MS as follows:

Although solar UV radiation reaching earth’s surface does not contain UVC, due to its absorption by atmospheric oxygen (wavelengths below 200 nm) and by the stratospheric ozone layer (absorbing wavelengths 200-290nm); solar UV radiation on earth surface has been proposed to be an environmental determinant in shaping COVID-19 transmission at the seasonal time scale [10] and it has been clearly negatively correlated with infection levels and mortality of patients [11-14].
